# Identification of a homology-independent linchpin domain controlling mouse and bank vole prion protein conversion

Cassandra M. Burke[1☯], Kenneth M. K. Mark[1☯], Daniel J. Walsh[1], Geoffrey P. Noble[1], Alexander D. Steele[1], Abigail B. Diack[2], Jean C. Manson[2], Joel C. Watts[3], Surachai Supattapone[1,4]*

1 Department of Biochemistry and Cell Biology, Geisel School of Medicine at Dartmouth, Hanover, New Hampshire, United States of America, 2 The Roslin Institute and R(D)SVS, University of Edinburgh, Easter Bush, Scotland, United Kingdom, 3 Tanz Centre for Research in Neurodegenerative Diseases, Department of Biochemistry, University of Toronto, Toronto, Ontario, Canada, 4 Department of Medicine, Geisel School of Medicine at Dartmouth, Hanover, New Hampshire, United States of America

☯ These authors contributed equally to this work.
* supattapone@dartmouth.edu

**Data Availability Statement:** All relevant data are within the manuscript and its Supporting Information files.

## Abstract

Prions are unorthodox pathogens that cause fatal neurodegenerative diseases in humans and other mammals. Prion propagation occurs through the self-templating of the pathogenic conformer PrP$^{Sc}$, onto the cell-expressed conformer, PrP$^C$. Here we study the conversion of PrP$^C$ to PrP$^{Sc}$ using a recombinant mouse PrP$^{Sc}$ conformer (mouse protein-only recPrP$^{Sc}$) as a unique tool that can convert bank vole but not mouse PrP$^C$ substrates *in vitro*. Thus, its templating ability is not dependent on sequence homology with the substrate. In the present study, we used chimeric bank vole/mouse PrP$^C$ substrates to systematically determine the domain that allows for conversion by Mo protein-only recPrP$^{Sc}$. Our results show that that either the presence of the bank vole amino acid residues E227 and S230 or the absence of the second N-linked glycan are sufficient to allow PrP$^C$ substrates to be converted by Mo protein-only recPrP$^{Sc}$ and several native infectious prion strains. We propose that residues 227 and 230 and the second glycan are part of a C-terminal domain that acts as a linchpin for bank vole and mouse prion conversion.

## Author summary

Prions are unconventional infectious agents that lack nucleic acids such as DNA and RNA, and the mechanism by which prions replicate is not fully understood. It has been established that a central feature of the replication mechanism involves the misfolding of a host protein (PrP$^C$) into an infectious shape termed PrP$^{Sc}$, but it is unclear how this misfolding occurs. Interestingly, it has been observed that a particular animal species, the European bank vole, is unusually susceptible to prion infection and that this near-universal susceptibility is caused by the specific PrP$^C$ sequence of this protein. Here we use a powerful and unique biochemical system to determine the specific region of bank vole

**Funding:** This work was supported by NIH grants R01NS102301, R01NS118796, R01NS117276 to S.S., NIH IDeA award to Dartmouth BioMT P20-GM113132, and NIH grant T32AI007519 to C.B. The funders had no role in study design, data collection and analysis, decision to publish, or preparation of the manuscript.

**Competing interests:** The authors have declared that no competing interests exist.

PrP$^C$ that is primarily responsible for its propensity to misfold into PrP$^{Sc}$. This critical region, which is located at the extreme C-terminal end of the protein, appears to act as a linchpin domain that normally stabilizes the shape of PrP$^C$ and thereby regulates its misfolding into PrP$^{Sc}$.

## Introduction

Prions are unorthodox infectious agents of fatal neurological diseases that affect many mammalian species, including humans. The central pathogenic event underlying prion infection is the self-propagating conformational change of a host-encoded glycoprotein (PrP$^C$) into a misfolded conformer (PrP$^{Sc}$)[1].

The protein-only hypothesis states that infectious prions are composed solely of PrP$^{Sc}$[2, 3]. However, much evidence indicates that cofactor molecules are required for the formation of infectious prions[4–7]. In direct support of this concept, our laboratory generated two self-propagating bacterially-expressed recombinant (rec) PrP$^{Sc}$ conformers (cofactor recPrP$^{Sc}$ and protein-only recPrP$^{Sc}$) by propagating the same original seed in the presence and absence of purified phospholipids, respectively[7, 8]. Whereas cofactor recPrP$^{Sc}$ effectively seeds mouse (Mo) brain homogenate (BH) sPMCA reactions *in vitro* and potently infects wild-type mice *in vivo*, protein-only recPrP$^{Sc}$ cannot seed Mo BH sPMCA reactions and fails to infect wild-type mice[7].

Most animal species have transmission barriers that render them resistant to the majority of prion strains from other species. Some species, such as rabbits, dogs, and horses are resistant to naturally occurring prion strains[9]. In contrast, the European bank vole (*Myodes glareolus)* is a highly susceptible host for a wide variety of prion diseases[10–14]. Experiments with transgenic mice showed that the susceptibility of bank voles to prion infection is ultimately encoded by the amino acid sequence of bank vole (BV) PrP$^C$[13, 15], and BV PrP has been shown to be a highly susceptible substrate for *in vitro* conversion assays [16]. Several residues and domains in the BV PrP$^C$ sequence have previously been identified through *in vitro* and *in vivo* approaches to play important roles in the susceptibility by specific prions from other species [17–22]. While the residues and domains identified in these experiments appear to play roles for the specific species barriers studied, it is difficult to distinguish whether they do so because of sequence mismatch between seed and substrate or because they control the structural transition of PrP$^C$ to PrP$^{Sc}$.

Recently, we discovered that protein-only recPrP$^{Sc}$ is able to potently seed BV but not Mo PrP$^C$ substrate in sPMCA reactions, and the PrP$^{Sc}$ molecules produced in these reactions are highly infectious[23]. Interestingly, we observed that the formation of infectious PrP$^{Sc}$ molecules from non-infectious protein-only recPrP$^{Sc}$ seed required several factors: (1) bank vole rather than mouse PrP$^C$ substrate, (2) native PrP$^C$ rather than recPrP substrate lacking post-translational modifications (PTMs), and (3) cofactor molecules.

In the present study, we exploit this system to identify PrP$^C$ domains that might serve as "linchpins" to control PrP conformational change. Uniquely, this system has the critical advantage of not being influenced by species barriers[24] since **mouse** protein-only recPrP$^{Sc}$ can seed BV PrP$^C$ **but not Mo PrP$^C$** in this system. Therefore, we can test the ability of native Mo-BV PrP$^C$ chimeric substrates to convert in sPMCA reactions driven by Mo protein-only recPrP$^{Sc}$ seed, with confidence that this process depends only upon the enhanced ability of BV PrP$^C$ (and chimeric substrates) to convert into PrP$^{Sc}$, rather than the degree of amino acid sequence similarity between seed and substrate. Similarly, this system also provides a unique

opportunity to study the effect of PTMs on PrP$^{Sc}$ formation since protein-only recPrP$^{Sc}$, which lacks PTMs, requires native PrP$^C$ substrate to produce PrP$^{Sc}$.

## Results

### BV PrP is uniquely able to propagate Mo protein-only recPrP$^{Sc}$

The amino acid sequence of the processed region of BV PrP is a natural chimera of Mo PrP and Ha PrP, with the exception of two unique residues at the extreme C-terminus (**Fig 1A**). As a preliminary experiment, we first compared the conversion ability of BV BH, Ha BH, and Mo BH to propagate Mo protein-only recPrP$^{Sc}$. The results show that BV BH, but not Ha BH or Mo BH, is capable of propagating Mo protein-only recPrP$^{Sc}$ (**Fig 1B, third column**). BV BH is

## A.

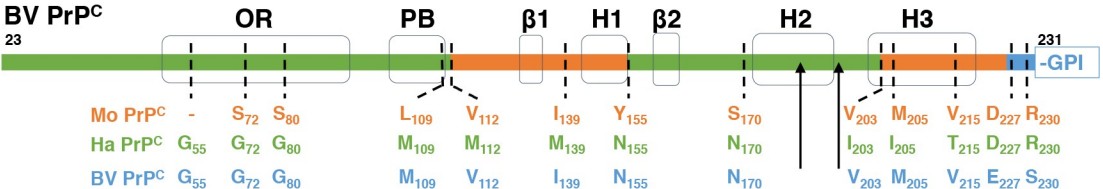

## B.

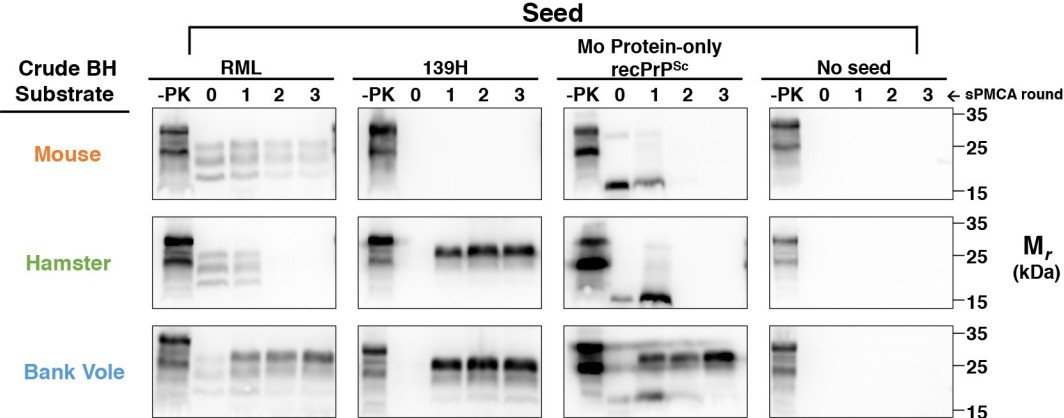

**Fig 1. Susceptibility of various rodent species in BH sPMCA. (A)** Amino acid comparison of the processed regions of Mo PrP, Ha PrP, and BV PrP. The sequence bar highlights the regions of BV PrP homologous to Mo PrP (orange) or Ha PrP (green), or are unique to BV PrP (blue). Black arrowheads denote the location of N-linked glycans. The locations of various structural domains are shown in black. OR = octapeptide repeat; PB = polybasic domain; GPI = glycophosphaditylinositol. Sequence alignments were performed using Multalin[25]. **(B)** Western blots showing three-round sPMCA reactions using normal brain homogenates (BH) from the species used in (A) as the substrates and initially seeded on day 0 with various seeds, as indicated. Day 0 samples are seeded reactions not subject to sonication. -PK = samples not subjected to proteinase K digestion; all other samples were proteolyzed.

also capable of propagating both the hamster prion strain 139H and the mouse prion strain RML. In contrast, Mo BH is capable of propagating only RML, and Ha BH is capable of propagating only 139H (**Fig 1B**). Taken together, these results show that BV PrP is a uniquely susceptible substrate to Mo protein-only recPrP$^{Sc}$ and native prion strains from other species.

We next developed a system that allowed us to design and produce a wide variety of PrP$^C$ substrates in HEK 293 cells. HEK 293 are human cells that can express secretory and membrane-bound proteins with post-translational modifications, including N-linked glycans and a GPI anchor[26, 27], at high level. All HEK-expressed constructs were partially deglycosylated using PNGase F as previously described (**S1 Fig**) to improve the conversion efficiency of Mo PrP$^C$[28] (**S2 Fig**). This effect is likely due to the previously observed inhibitory effects of PrPC glycosylation on the propagation of mouse prion strains[28, 29], which may be due to negatively charged sialic acid groups within the glycan structure[30], and which may be further aggravated by hyper-glycosylation in HEK 293 cells. To ensure our system faithfully reproduces similar seed specificity as brain homogenate substrates in sPMCA, we first tested the susceptibility of cell-expressed BV PrP and Mo PrP in sPMCA (**Fig 2**). The susceptibility of each construct was tested in three-round reconstituted BH sPMCA reactions containing PrP$^{0/0}$ brain homogenate and a partially purified PrP$^c$ construct. The results show that, like the crude BH substrates, cell-expressed BV PrP$^C$ is capable of propagating Mo protein-only recPrP$^{Sc}$, 139H, and RML (**Fig 2, top row**). Cell-expressed BV PrP$^C$ can also propagate Mo cofactor recPrP$^{Sc}$. However, significant PrP$^{Sc}$ formation was not observed in either the 139H-seeded or Mo cofactor recPrP$^{Sc}$ seeded lanes until round 3 of sPMCA (**Fig 2, top row, second and third panels**). In contrast, cell-expressed Mo PrP$^C$ was capable of propagating RML and Mo cofactor recPrP$^{Sc}$, but not Mo protein-only recPrP$^{Sc}$ or 139H (**Fig 2, bottom row**). Taken together, the results show that the cell-expressed PrP$^C$ substrates have similar susceptibilities to crude BH substrates from the same species.

## BV PrP C-terminal domain is required for efficient conversion of sPMCA reactions seeded with Mo protein-only recPrP$^{Sc}$

To identify the specific amino acid residue(s) required to enable BV PrP$^C$ to propagate Mo protein-only recPrP$^{Sc}$, we designed a series of chimeric constructs based on the BV PrP$^C$ backbone sequence that, working from the C- towards the N- terminus, become progressively substituted with Mo PrP residues (**Fig 3A**), and tested the ability of these chimeric constructs to be seeded by Mo protein-only recPrP$^{Sc}$ in sPMCA reactions. Within this systematic

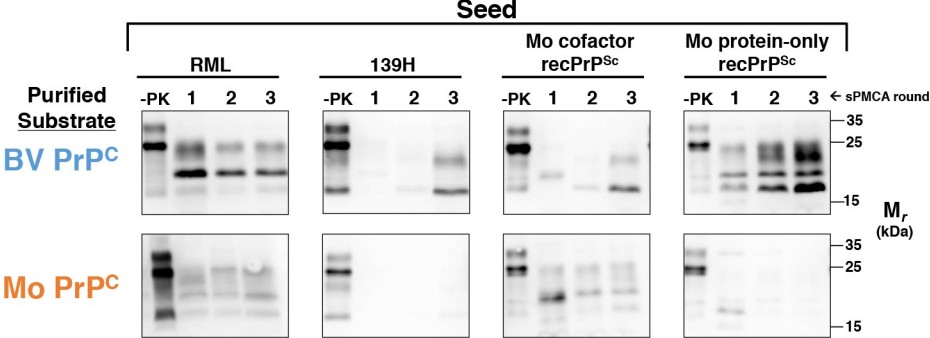

**Fig 2. Susceptibility of BV PrP$^C$ and Mo PrP$^C$ substrates in reconstituted sPMCA.** Western blots showing three-round reconstituted sPMCA reactions using either BV or Mo partially purified PrP$^C$ substrates supplemented with PrP$^{0/0}$ BH and seeded with various seeds, as indicated. -PK = samples not subjected to proteinase K digestion; all other samples were proteolyzed.

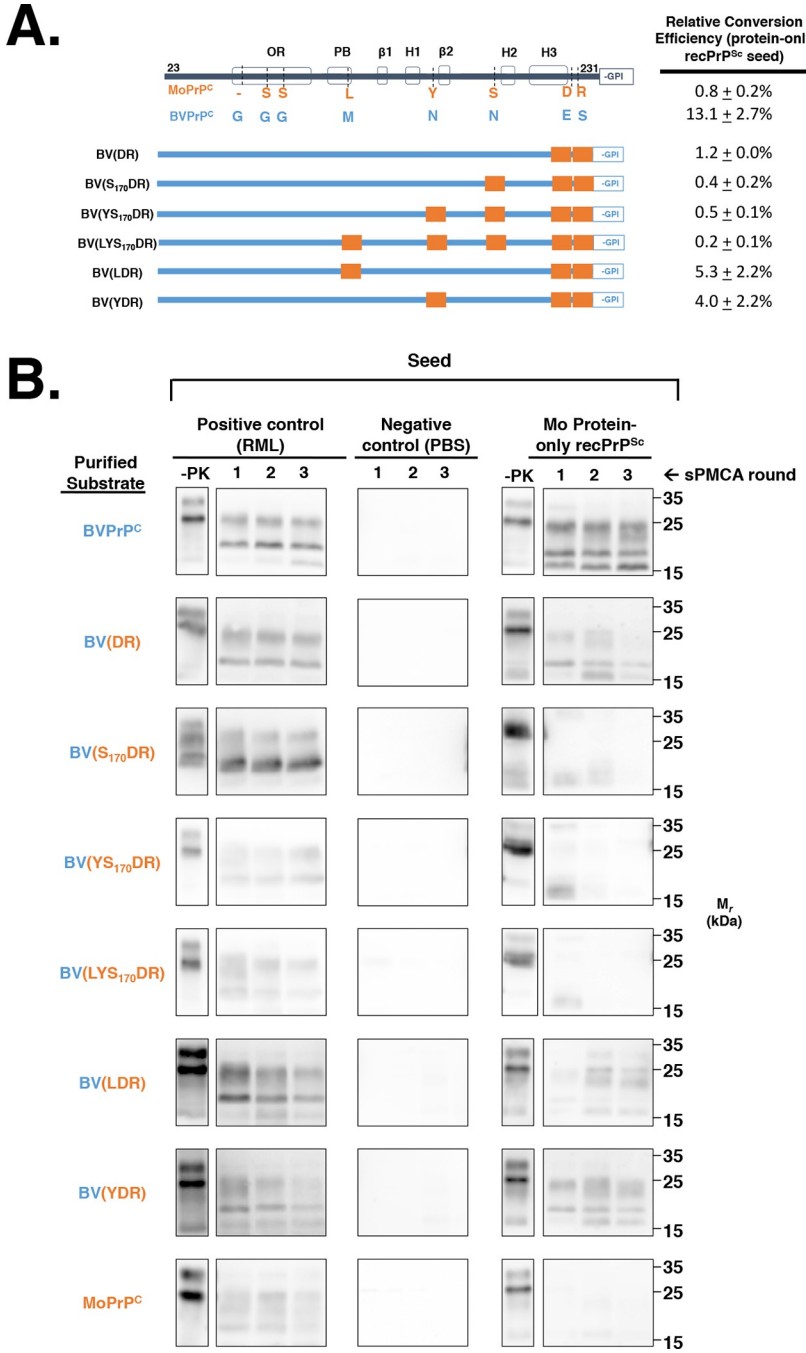

**Fig 3. Determining the minimum PrP sequence required for propagation of Mo protein-only recPrP^Sc seed. (A)** Table summarizing the ability of various BV/Mo PrP^C chimeras to propagate Mo protein-only recPrP^Sc. All constructs in this series are based on the BV PrP^C backbone and become progressively substituted with Mo residues from the C-terminus towards the N-terminus, except for the last two constructs which contain different combinations of 3 Mo substitutions. **(B)** Western blots of experiments summarized in (A), showing three-round reconstituted sPMCA reactions using partially purified PrP^C substrates supplemented with PrP^{0/0} BH and seeded with various seeds, as indicated. -PK = samples not subjected to proteinase K digestion; all other samples were proteolyzed.

paradigm, BV PrP substitution(s) that are specifically required to propagate Mo protein-only recPrP^Sc are identified when the resulting chimera is no longer capable of propagating Mo protein-only recPrP^Sc. The results show that substitution of residues 227 and 230 from BV

sequence to Mo sequence (i.e. E227D and S230R) inhibited conversion of Mo protein-only recPrP$^{Sc}$ seeded reactions by >10-fold (**Fig 3A, third row), chimera BV(DR)**). We observed some variability in sPMCA patterns obtained with BV(DR) substrate from experiment-to-experiment; we detected no bands at all in round three in 3/5 Mo protein-only recPrP$^{Sc}$-seeded independent sPMCA experiments. In other cases, such as the example shown (**Fig 3B, second row right panel**), we observed a near-complete absence of the ~24 kDa PK-resistant by round three of the reaction, while the lower molecular weight bands at ~16 and 19 kDa are present in round three, but have reduced signal intensity. In comparison, the positive control reaction containing BV(DR) PrP$^C$ and seeded with RML PrP$^{Sc}$ showed consistent three-round propagation (**Fig 3B, second row, left panel**).

BV(DR) chimeras with additional step-wise N-terminal Mo substitutions (i.e. BV(S$_{170}$DR), BV(YS$_{170}$DR) and BV(LYS$_{170}$DR) displayed no significant conversion in reactions seeded with Mo protein-only recPrP$^{Sc}$ (**Fig 3A, rows 4–6; and Fig 3B, rows 3–5, right panel**). In contrast, BV(LDR) and BV(YDR) substrates were able to propagate Mo protein-only recPrP$^{Sc}$ at ~1/3 the conversion efficiency of BV PrP$^C$ (**Fig 3A, rows 7–8; and Fig 3B, rows 6–7, right panel**). Taken together, our results indicate that residues E227 and S230 are required for efficient conversion of sPMCA reactions seeded by Mo protein-only recPrP$^{Sc}$, and additional substitutions of other non-homologous residues also influence the degree of conversion efficiency.

Finally, we also designed and tested the ability of a series of chimeric constructs, which substituted single Mo PrP amino acid residues to their cognate BV PrP residue, to propagate Mo protein-only recPrP$^{Sc}$ (**Fig 4A and Fig 4B**). This analysis showed that neither E227D nor S230R substitution alone reduced the conversion efficiency of BV PrP$^C$ (**Fig 4A, rows 3–4; and Fig 4B, rows 1–2, right panel**). On the other hand, BV(S$_{170}$) and BV(L) displayed lower conversion efficiencies than BV PrP$^C$ (**Fig 4A, rows 5 and 7; and Fig 4B, rows 3 and 5, right panel**) These data indicate that the E227D and S230R substitutions work synergistically to block propagation of Mo protein-only recPrP$^{Sc}$, whereas substitution of other individual residues can reduce the conversion efficiency of chimeric PrP$^C$ molecules.

## The extreme C-terminal domain of BV PrP$^C$ is sufficient to propagate Mo protein-only recPrP$^{Sc}$

We next sought to identify bank vole amino acid residue(s) that are sufficient to propagate Mo protein-only recPrP$^{Sc}$. Therefore, we designed a series of chimeric constructs based on the Mo PrP$^C$ backbone sequence that, working from the C- to the N-terminus, become progressively substituted with BV PrP residues (**Fig 5A**). and tested the ability of these chimeric constructs to be seeded by Mo protein-only recPrP$^{Sc}$ in sPMCA reactions. The results show that substitution of the two non-homologous residues in extreme C-terminus (227 and 230) from DR to ES is sufficient to propagate Mo protein-only recPrP$^{Sc}$ (**Fig 5B, first row; Mo(ES), right panel**).

We also designed and tested the ability of a series of chimeric constructs that substitute single BV PrP amino acid residues to their cognate Mo PrP residue to propagate Mo protein-only recPrP$^{Sc}$ (**Fig 6A**). The results show that this series of chimeric PrP$^C$ substrates are unable to propagate Mo protein-only recPrP$^{Sc}$ for three rounds in sPMCA (**Fig 6B, right panels**). In each case, we observed a decrease in PK-resistant PrP$^{Sc}$ over each round of sPMCA, resulting in the near absence of PK-resistant PrP$^{Sc}$ by round three in each reaction tested (**Fig 6B, right panels**). In contrast, we found that the positive control RML PrP$^{Sc}$ efficiently propagates in reactions this series of chimeric PrP$^C$ substrates for three rounds of sPMCA (**Fig 6B, left panels**). Thus, no single amino acid mutation from Mo to BV PrP$^C$ sequence is sufficient to restore the ability to propagate Mo protein-only recPrP$^{Sc}$. Overall, our results show that both the E227D and S230R substitutions are required to enable Mo PrP$^C$ chimeras to propagate Mo protein-only recPrP$^{Sc}$.

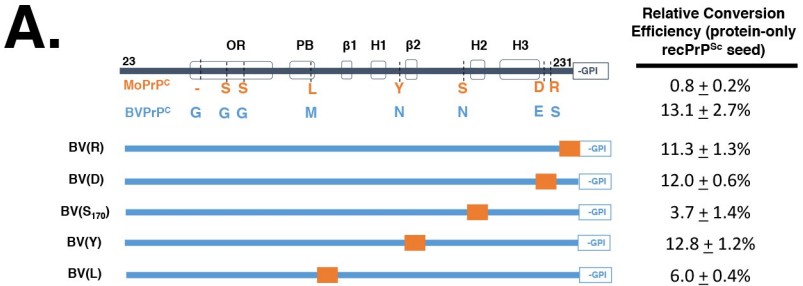

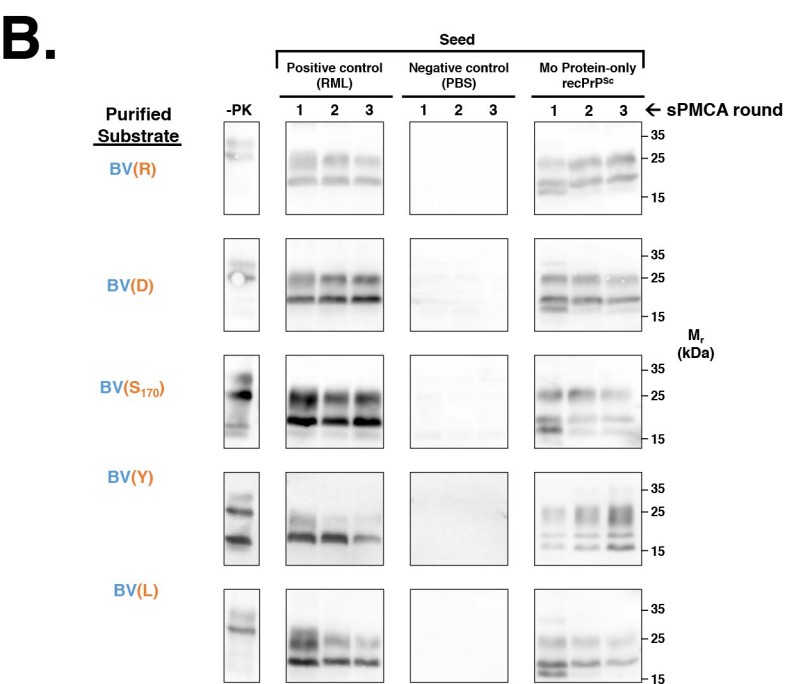

**Fig 4. Determining the effect of single Mo amino acid mutations on the propagation of Mo protein-only recPrP$^{Sc}$.**
(**A**) Table summarizing the effect of single Mo amino acid substitutions on the ability of BV/Mo PrP$^{C}$ chimeras to propagate Mo protein-only recPrP$^{Sc}$. All constructs are based on the BV PrP$^{C}$ backbone. (**B**) Western blots of experiments summarized in (A), showing three-round reconstituted sPMCA reactions using partially purified PrP$^{C}$ substrates supplemented with PrP$^{0/0}$ BH and seeded with various seeds, as indicated. -PK = samples not subjected to proteinase K digestion; all other samples were proteolyzed.

## The extreme C-terminus also controls propagation of native infectious prion strains

To evaluate whether 227 and 230 also controls the ability of PrP$^{C}$ substrate to propagate native infectious prion strains, we tested the ability of hamster Sc237, sheep scrapie, and deer CWD to seed the critical chimeric substrates affecting residues 227 and 230, BV(DR) and Mo(ES). Like protein-only recPrP$^{Sc}$, all three of these native strains can seed BV but not Mo PrP$^{C}$ substrate (**Fig 7, top and third rows**) (. Unlike BV PrP$^{C}$, the BV(DR) chimera was unable to propagate Sc237, sheep scrapie, or CWD (**Fig 7, second row**). Conversely, unlike Mo PrP$^{C}$, the Mo (ES) chimera was competent substrate for propagation of these three native strains (**Fig 7, bottom row**). As expected, all of the wild-type and chimeric substrates could successfully propagate mouse RML, a strain that infects both mice and bank voles efficiently (**Fig 7, first seed**

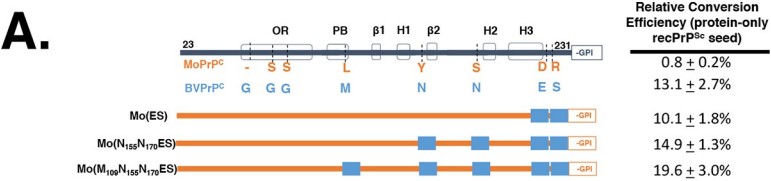

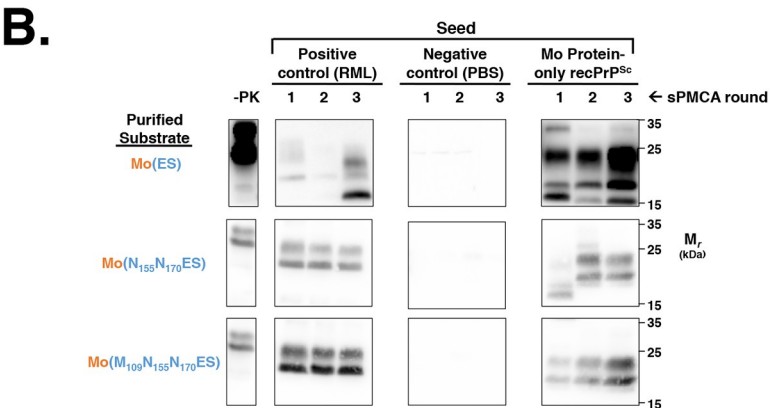

**Fig 5. Determining the BV amino acids that are sufficient to propagate protein-only recPrP^Sc. (A)**
Table summarizing the ability of BV/Mo PrP^C chimeras to propagate Mo protein-only recPrP^Sc. All constructs are
based on the Mo PrP^C backbone and become progressively substituted with BV residues from the C-terminus towards
the N-terminus. **(B)** Western blots of experiments summarized in (A), showing three-round reconstituted sPMCA
reactions using partially purified PrP^C substrates supplemented with PrP^{0/0} BH and initially seeded with various seeds,
as indicated. -PK = samples not subjected to proteinase K digestion; all other samples were proteolyzed.

**column).** Taken together, these results indicate that residues 227 and 230 play a critical role in
determining the enhanced susceptibility of BV PrP^C to seeding by a variety of native prion
strains.

In some reactions, we observed changes in the migration pattern of PK-resistant bands
between sPMCA rounds (**Fig 7, RML- and sheep scrapie-seeded BVPrP^C and CWD-seeded
Mo(ES))** These shifts may represent strain adaptation, which can occur rapidly and stochasti-
cally during sPMCA propagation [31], and which is more likely to occur during cross-species
propagation[32].

## Specific PrP^C glycosylation mutant is also able to propagate Mo protein-only recPrP^Sc

Finally, we also sought to investigate the influence of PTMs (rather than amino acid sequence)
on PrP^C susceptibility to Mo protein-only recPrP^Sc. To do this, we used brain homogenates
from transgenic mice expressing various mutant Mo PrP^C molecules with specific PTM defects
as substrates in sPMCA reactions. We first analyzed a series of previously described glycosyla-
tion mutants with known differences in susceptibility to infection by various prion strains *in
vivo*[33]. Within this series, G1 mice express PrP^C with a point mutation at the first N-linked
glycosylation site, G2 mice express PrP^C with a point mutation at the second N-linked glyco-
sylation site, and G3 mice express PrP^C containing both mutations[34, 35]. We carried out
sPMCA experiments using G1, G2, or G3 brain homogenate substrate seeded with Mo pro-
tein-only recPrP^Sc or RML. Remarkably, the results show that G2 PrP^C was able to convert
into PrP^Sc when seeded by Mo protein-only recPrP^Sc (**Fig 8, middle panel**). The conversion

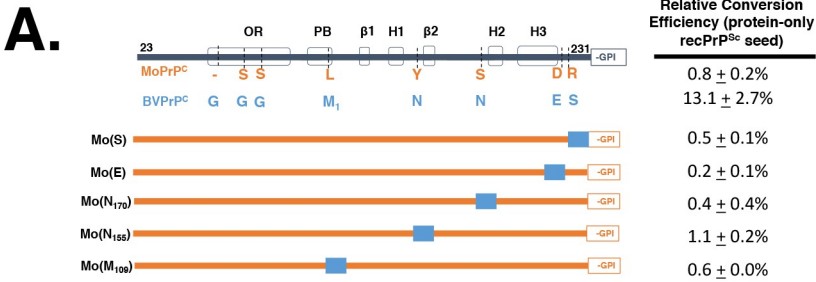

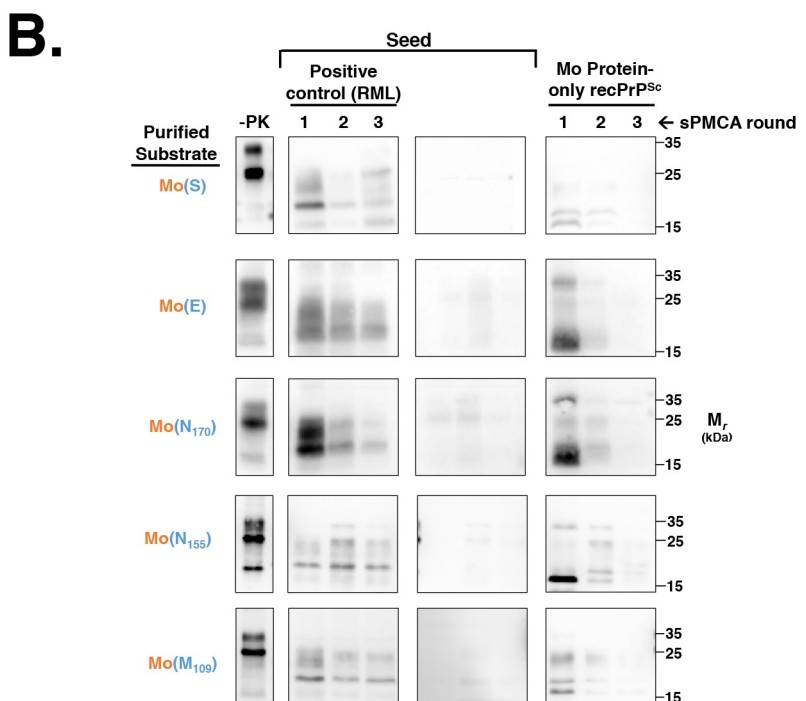

**Fig 6. Determining the effect of single BV amino acid mutations on the propagation of Mo protein-only recPrP^Sc.** (**A**) Table summarizing the effect of single BV amino acid substitutions on the ability of BV/Mo PrP^C chimeras to propagate Mo protein-only recPrP^Sc. All constructs are based on the Mo PrP^C backbone. (**B**) Western blots of experiments summarized in (A), showing three-round reconstituted sPMCA reactions using partially purified PrP^C substrates supplemented with PrP^0/0 BH and initially seeded with various seeds, as indicated. -PK = samples not subjected to proteinase K digestion; all other samples were proteolyzed.

process appears to be slower for G2 PrP^C than for susceptible chimeric PrP^C substrates, since G2 PrP^Sc became detectable only after 4 rounds of sPMCA propagation. In contrast, neither G1 nor G3 PrP^C appeared to be susceptible to seeding by protein-only recPrP^Sc (**Fig 8, top and bottom panels**).

## Discussion

Here we leveraged a unique *in vitro* conversion system to identify a specific domain within the prion protein (containing residues 227 and 230) that appears to serve as a linchpin that controls the conformational change of PrP^C to PrP^Sc. The amino acid sequences of mature bank vole PrP^C and mouse PrP^C differ at only 8 residues[12, 13], and our system allowed us to systematically examine which of these non-homologous amino-acid residues are responsible for the remarkable susceptibility of BV PrP^C to convert into PrP^Sc. For the purposes of this study,

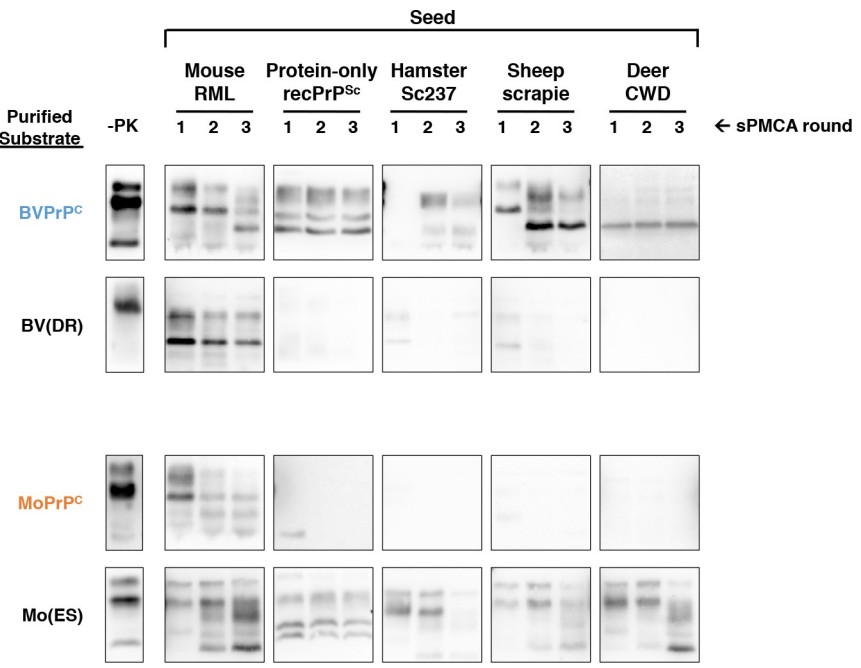

**Fig 7. Determining the effect of specific BV and Mo amino acids on the species barrier.** Western blots of experiments showing three-round reconstituted sPMCA reactions using partially purified PrP$^C$ substrates supplemented with PrP$^{0/0}$ BH and initially seeded on day 0 with various seeds, as indicated. Day 0 samples are a seeded reaction not subject to sonication. -PK = samples not subjected to proteinase K digestion; all other samples were proteolyzed.

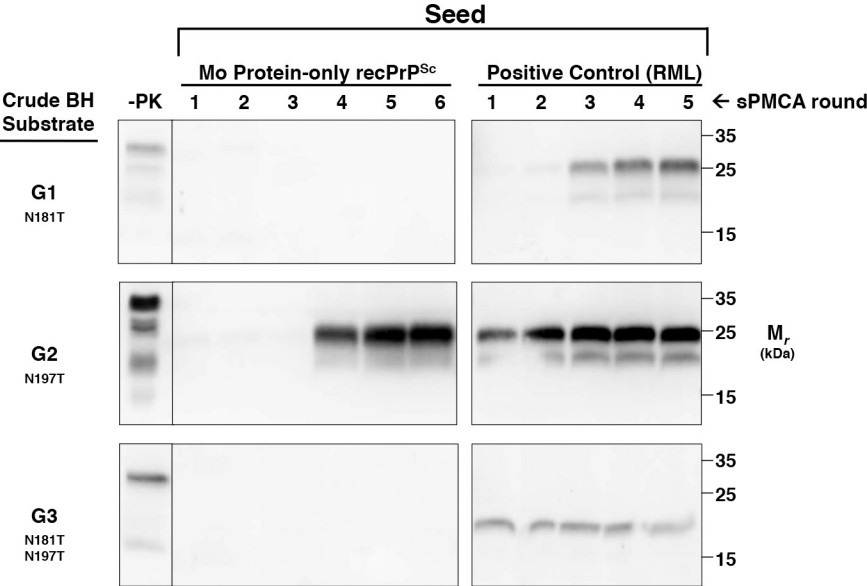

**Fig 8. Propagation of Mo protein-only recPrP$^{Sc}$ in sPMCA reactions with glycosylation-deficient PrP$^C$ substrates.** Western blots of sPMCA reactions using brain homogenates from glycosylation mutants G1, G2, or G3 as substrates, as indicated. Reactions were seeded with Mo protein-only recPrP$^{Sc}$ or RML, as indicated. -PK = samples not subjected to proteinase K digestion; all other samples were proteolyzed. Mouse amino acid numbering scheme is used to show locations of N-linked glycans on Mo PrP.

our system provided two critical technical advantages. (1) The ability of Mo protein-only recPrP$^{Sc}$ to convert BV PrP$^C$ does not depend on amino acid complementarity between seed and substrate, allowing us to interpret our results without the potentially confounding factor of sequence mismatch. (2) Our cell-based expression system allowed us to produce multiple BV-Mo chimeric PrP$^C$ constructs rapidly and inexpensively, allowing us to perform a thorough analysis of potential permutations in an unbiased manner.

## Identification of a C-terminal "linchpin" domain that controls prion protein conformational change

The major finding of this study is that the propensity of BV PrP$^C$ to undergo conformational change to PrP$^{Sc}$ appears to be primarily controlled by two residues (227 and 230) located within the extreme C-terminus of mature PrP. Mutation of these two residues within the Mo PrP$^C$ backbone to their corresponding BV sequence identities was sufficient to allow seeded conversion of the resulting chimera by Mo protein-only recPrP$^{Sc}$. Conversely, mutation of these same two residues within the BV PrP$^C$ backbone to their corresponding Mo sequence identities largely inhibited conversion seeded by protein-only recPrP$^{Sc}$. Based on these findings, we propose that the extreme C-terminal domain of PrP$^C$ may serve as a key "linchpin" for its conformational change. We specifically infer that the E227 and S230 residues destabilize this domain within BV PrP$^C$ to facilitate its conformational change into a variety of PrP$^{Sc}$ conformers and prion strains. In support of this concept, we found that these two amino acids also control susceptibility to hamster Sc237, sheep scrapie, and deer CWD prion strains. Notably, Kobayashi *et al.* reported that ~40% of transgenic mice overexpressing Mo PrP$^C$ containing the E227 and S230 mutations develop a spontaneous prion disease [36]. Taken together, the results of our systematic analysis using seeded conversion assays and the spontaneous disease reported by Kobayashi *et al.* provide complementary and compelling evidence for the hypothesis that the extreme C-terminus is a linchpin domain, which controls the conversion (either seeded or unseeded) of PrP$^C$ into PrP$^{Sc}$.

It is worth noting that, among the 8 BV residues that do not have homologous match in the Mo PrP sequence, E227 and S230 are unique to vole species (**Fig 1**). In contrast, the other 6 mismatched BV residues can be found in other mammalian species; for instance, all 6 are homologous to the corresponding residues within the hamster (Ha) PrP sequence (**Fig 1**). Together, the observations that (1) residues E227 and S230 are unique to vole species and (2) voles are uniquely susceptible to prion diseases[10–14] are consistent with the hypothesis that the extreme C-terminus plays a key role in prion formation, possibly by stabilizing PrP$^C$. This hypothesis is also consistent with a number of previous results. (1) This region displays decreased solvent accessibility upon transition from PrP$^C$ to PrP$^{Sc}$ [37], indicating it undergoes conformational change. (2) Residues 225 and 226 of deer PrP$^C$ (222 and 223 of BV PrP$^C$) appear to play critical roles in interspecies prion conversion [38]. (3) Specific polymorphisms and mutations in the C-terminus can produce dominant negative PrP$^C$ molecules [39]. (4) Y145stop, a PrP mutant lacking the C-terminus, appears to convert spontaneously into amyloid fibers[40] (the absence of the C-terminus in this truncated mutant may destabilize PrP structure and promote misfolding). Finally, it is interesting to note that the GPI anchor is attached to the C-terminus of PrP. It is possible that attachment to this bulky modification may normally stabilize the C-terminal domain to prevent the conformational change of native PrP$^C$ into PrP$^{Sc}$. Consistent with this notion, transgenic mice that overexpress Mo PrP$^C$ lacking a GPI anchor spontaneously develop prion disease[41].

It is important to acknowledge the experimental limitations of our work. (1) We have not confirmed that the various chimeric PrP$^{Sc}$ molecules produced in our sPMCA reactions are

infectious. However, we have previously shown that similar reactions containing BV PrP$^C$ substrate and seeded with protein-only recPrP$^{Sc}$ produces fully infectious prions[23], making it likely that we are also studying a process relevant to infectious prion formation in this study. (2) Since we are using Mo protein-only recPrP$^{Sc}$ as a tool to study prion conversion, we can only say that our results are sequence-independent, and may not hold for every prion strain (we have only been able to test the few native strains that show differential ability to seed BV vs. Mo PrP$^C$ substrate in reconstituted sPMCA reactions). Additional *in vitro* and *in vivo* studies are needed to fully characterize the role played by the C-terminal domain in the propagation of naturally occurring prion strains (3) Finally, our PrP$^C$ substrates are partially deglycosylated, and the glycosylation status of PrP$^C$ has been shown to modulate its susceptibility to prion infection *in vivo* [35, 42], raising the possibility that the deglycosylation step might artifactually change the results of our chimeric analysis. However, this appears to be unlikely because we found that our partially deglycosylated WT BV PrP$^C$ and Mo PrP$^C$ substrates display similar patterns of seeding specificity as their fully glycosylated counterparts in sPMCA reactions.

## The influence of other non-homologous residues on seeded conversion by protein-only recPrP$^{Sc}$

In addition to identifying residues 227 and 230 as a linchpin domain, our work also showed that other non-homologous residues could influence the ability of chimeric molecules to propagate Mo protein-only recPrP$^{Sc}$. Specifically, we observed that additional substitution of either M109L or N155Y partially restored the conversion ability of the BV(DR) chimera (**Fig 3A, rows 7–8**), whereas additional substitution with N170S inhibited conversion of BV(DR) (**Fig 3A row 4**) as well as BV PrP$^C$ (**Fig 4A, row 5**). The former effects are expected outcomes of increasing homology between substrate and seed in non-linchpin domains, but the latter effect at residue 170 appears paradoxical, since increasing homology to the MoPrP sequence actually reduces the ability of chimeric substrates to propagate Mo protein-only recPrP$^{Sc}$.

It has been previously reported that homology between PrP$^C$ and PrP$^{Sc}$ at position 170 correlates with interspecies prion conversion *in vitro*[43] and is important in allowing transmission across the species barrier *in vivo*[44]. Additionally, residues 155 and 170 have been shown to control the conversion efficiency of BVPrP$^C$ in cell-free conversion (CFC) assays *in vitro* [19] and control susceptibility to prion infection in rodents[17]. Surprisingly, our results paradoxically show that increasing homology at position 170 between PrP$^C$ and PrP$^{Sc}$ blocks conversion with Mo protein-only recPrP$^{Sc}$. PrP forms a β2- α2 loop in the region 165–171[20], and mutations that modify this structure can alter host susceptibility[45]. Mice, which have serine at position 170, have a disordered loop [46, 47], while bank voles which have asparagine at position 170 have a rigid loop [20]. Our data supports a model whereby structural or biophysical elements dictated by the amino acid sequence of PrP$^C$ that correlate with loop rigidity, rather than amino acid sequence homology between PrP$^C$ and PrP$^{Sc}$, promote the conversion of PrP$^C$ to PrP$^{Sc}$. This model is supported by our finding that a mutation that increases loop homology but decreases rigidity blocks conversion by Mo protein-only recPrP$^{Sc}$ together with a series of observations from other investigators. (1) Transgenic mice that 2-3x overexpress PrP containing 2 point mutations (170N,174T) that increase the rigidity of the a β2- α2 loop develop a spontaneous prion disease[48]. (2) The 170N, 174T double mutation was also found to increase the propensity of recPrP to form amyloid fibrils *in vitro*[49]. (3) Transgenic mice overexpressing 3-5x PrP containing another mutation that increases the rigidity of the β2- α2 loop (D167S) develop a spontaneous neurologic disease [50].

It is worth noting that the impact of 170 substitution on PrP conversion is asymmetrical. In other words, the inverse substitution (S170N) into the Mo PrP$^C$ backbone does not produce a

chimera that is capable of propagating Mo protein-only recPrP$^{Sc}$ (**Fig 6A, row 5**). We can therefore infer that the extreme C-terminal domain likely plays a larger role than residue 170 in facilitating propagation of Mo protein-only recPrP$^{Sc}$.

## PrP$^C$ glycosylation adjacent to C-terminus also inhibits prion conversion

In addition to the amino acid sequence of the extreme C-terminus, we found that the propensity of PrP$^C$ to undergo prion conversion could also be affected by the adjacent N-linked glycan. Specifically, we found that mutant Mo PrP$^C$ lacking the second N-linked glycan (G2) at position 196 (mouse numbering) near the C-terminus can also propagate mouse protein-only recPrP$^{Sc}$ *in vitro*. The effect of G2 deglycosylation does not appear to be as important as the effect of the E227 and S230 mutations since G2 PrP$^{Sc}$ formation was not observed until round 4. In contrast, mutant PrP$^C$ lacking the other (G1), or both (G3) N-linked glycans remained resistant to conversion. It is surprising that Mo G2 PrP$^C$ should be more susceptible than Mo G3 PrP$^C$ to seeding by Mo protein-only recPrP$^{Sc}$, since Mo G3 PrP$^C$ is more similar to recPrP (in that both proteins are completely devoid of N-linked glycans). However, in line with our results, it has been previously reported that transgenic mice expressing Mo G2 PrP$^C$ are more susceptible than G1, G3, and wild-type mice to cross-species infection with human prions[33], suggesting that the first N-linked glycan may serve to facilitate PrP$^{Sc}$ formation for several different prion strains. However, the first N-linked glycan of PrP$^C$ may not be important for propagating all prion strains (e.g. Kim *et al.* produced a synthetic prion strain is infectious to mice expressing PrP$^C$ lacking both glycans[42]).

## Conclusion

We have leveraged a unique *in vitro* conversion system to show that the extreme C-terminus serves as a linchpin to control the seeded conversion of PrP$^C$ to PrP$^{Sc}$. Specifically, our data indicate that BV-specific residues at positions 227 and 230 are sufficient to promote prion conversion independent of seed/substrate sequence homology. We also found that the second N-linked glycan (adjacent to the C-terminal domain) inhibits seeding by Mo protein-only recPrP$^{Sc}$. While PrP$^C$ from both bank voles and G2 mutant mice are susceptible to conversion by recombinant protein-only protein seeds and other natural strains, both are relatively resistant to BSE-derived prions [17, 35], which is surprising since BSE prions readily infect wild-type mice[51] and many other animal species. Taken together, these observations suggest that Mo G2 PrP$^C$ and BV PrP$^C$ may share a common folding mechanism to form PrP$^{Sc}$, and that this mechanism may be able to accommodate a wide variety of PrP$^{Sc}$ conformations (except for BSE, which may propagate by a different folding mechanism). It is possible that the G2 glycan, which is attached to residue N196, normally serves to stabilize the C-terminal domain of PrP$^C$, and that specific residues perform a similar function, such as D227 and R230 in mouse PrP. Given that this linchpin controls the conversion of PrP$^C$ into PrP$^{Sc}$, we propose that stabilization of this region potentially represents an attractive target for potential prion therapeutics.

## Materials and methods

### Ethics statement

The Guide for the Care and Use of Laboratory Animals of the National Research Council was strictly followed for all animal experiments. All experiments conducted at Dartmouth College involving voles and mice in this study were conducted in accordance with protocol supa.su.1 as reviewed and approved by Dartmouth College's Institutional Animal Care and Use

Committee, operating under the regulations/guidelines of the NIH Office of Laboratory Animal Welfare (assurance number A3259-01) and the United States Department of Agriculture. All experiments conducted at the Roslin Institute involving mice were approved by The Roslin Institute's Animal Welfare Ethical Review Board (internal protocol number; A471) and were conducted according to the regulations of the 1986 United Kingdom Home Office Animals (Scientific Procedures) Act.

## Native prion strains

The mouse prion strain RML and the hamster prion strains 139H and sc237 were gifts from Stanley Prusiner (University of California, San Francisco, USA). The deer CWD prion strain and sheep scrapie prion strain were gifts from Mark Hall (USDA, Ames, Iowa, USA). The hamster strain Hyper was a gift from Suzette Priola (Rocky Mountain Laboratories, Hamilton, MT).

## General sPMCA methods

The general sPMCA experimental method was adapted from Castilla et al.[52]. All PMCA reactions were sonicated in microplate horns at 37°C using a Misonix S-4000 power supply (Qsonica, Newtown, CT) set to power 70 for three rounds. One round of PMCA is equal to 24 hr. The first round of PMCA was seeded with a volume of PrP$^{Sc}$ equal to 10% of the total reaction volume. RML prion-infected brain homogenates used as PrP$^{Sc}$ seeds were used at a final reaction concentration of 1.0% (v/v), and 139H prion-infected brain homogenates used as PrP$^{Sc}$ seeds were used at a final reaction concentration of 0.1% (v/v). To propagate the reaction between PMCA rounds, 10% of the reaction volume was transferred into a new, unseeded, substrate mixture. Due to the sensitivity of sPMCA[53], measures were undertaken to prevent sample contamination. Sample conical tubes were sealed with Parafilm (Bemis Company, Oshkosh, WI) and the sonicator horn was soaked in 100% bleach between experiments to prevent cross-contamination. The experimenter wore two pairs of gloves and changed the outer layer of gloves when handling a new sample. Sample conical tubes were spun at 500 $x$ $g$ for 5 sec to remove liquid off the conical tube lids before propagation and samples were propagated individually using aerosol resistant pipette tips. With each experiment, a sentinel conical tube (a conical tube containing the entire sPMCA reaction mixture but lacking seed) was also placed in the sonicator horn to monitor reactions for contamination.

## Preparation of cofactor and protein-only recPrP$^{Sc}$ by sPMCA

RecPrP, purified as described by Breydo et al.[54] was used to generate cofactor recPrP$^{Sc}$ and protein-only recPrP$^{Sc}$, as described[7, 23]. Briefly, 200 μL reactions containing 6 μg/mL recombinant mouse PrP 23–230 or M109 bank vole PrP 23–231 in conversion buffer [20 mM Tris pH 7.5, 135 mM NaCl, 5 mM EDTA pH 7.5, 0.15% Triton X-100] were supplemented with either brain-derived cofactor[7] for cofactor recPrP$^{Sc}$ propagation, or water for protein-only recPrP$^{Sc}$ propagation. Reactions were seeded with 20 μL of converted cofactor recPrP$^{Sc}$ or protein-only recPrP$^{Sc}$ and sonicated with 15 sec pulses every 30 min for 24 hr at 37°C.

## sPMCA with brain homogenate

A 10% (w/v) brain homogenate was prepared initially by Potter homogenization in PBS. The crude homogenate was spun at 400 x $g$ for 1 min, and the supernatant was removed and kept. Triton X-100 was added to the supernatant for a final concentration of 1% (v/v), and the supernatant was solubilized on ice for 10 min. Brains were taken from European bank voles with the

M109 genotype, Syrian hamsters, C57BL/6J mice, PrP$^{0/0}$ mice or transgenic mice expressing glycosylation mutants (G1, G2, G3) as previously described[35]. One-hundred microliter reactions were seeded with 10 μL of PrP$^{Sc}$ and sonicated with 20 sec pulses every 30 min for 24 hr at 37˚C.

### Reconstituted sPMCA with HEK 293-expressed PrP

Fifty-five microliters of HEK expressed PrP$^C$ substrate was mixed with 10 μL of 10X cell PMCA buffer (1% Triton X-100, 150 mM NaCl, 500 mM imidazole pH 7.5, 50 mM EDTA pH 7.5), 25 μL of 10% PrP$^{0/0}$ brain homogenate, and seeded with 10 μL of a sPMCA reaction or 10% brain homogenate. One-hundred microliter reactions were seeded with 10 μL of PrP$^{Sc}$ and sonicated with 20 sec pulses every 30 min for 24 hr at 37˚C. The final concentration of PrP$^C$ in each reaction was 4–5 μg/mL.

### Detection of PrP$^{Sc}$ in sPMCA reactions

Formation of PrP$^{Sc}$ was monitored by digestion of PMCA samples with proteinase K (PK) and western blotting. Samples were digested with 64 μg/mL PK (Roche, Basel, Switzerland) at 37˚C with shaking at 750 r.p.m. Samples from sPMCA reactions using recPrP as the substrate were treated for 30 min, while samples using HEK 293 expressed PrP or brain homogenate as the substrate were treated for 60 min. Digestions were quenched by adding SDS-PAGE loading buffer and heating to 95˚C for 15 min. SDS-PAGE and western blotting were performed as described previously [7] using mAb 27/33. Twenty microliters of a sPMCA reaction was subjected to PK digestion. The minus PK (-PK) lane is used to determine the conversion efficiency of a sPMCA reaction. For reactions using recombinant PrP or crude BH as the substrate, the -PK lane contains the same volume (20 μL) of a sPMCA reaction as a PK-digested sample. For reactions using HEK 293 expressed PrP the -PK lane contains a volume (2 μL) equivalent to one-tenth used in the PK-digested samples. This is because these substrates result in a much lower expected conversion efficiency. Images were quantified using Image Studio Lite (Licor) software to obtain the background-subtracted average signal intensity for the -PK and rounds 1–3 bands. The -PK value was multiplied by 10 to account for loading differences between it and rounds 1–3. The percent-conversion was then calculated for round 3 by dividing the background-subtracted average signal intensity for each round by the background-subtracted average signal intensity for the -PK round. The average percent conversion ±SEM for round 3 was calculated using between 1 and 5 biological replicates and 1 and 3 technical replicates.

We also confirmed that the final round PrP$^{Sc}$ molecules produced in sPMCA reactions using HEK-expressed PrP$^C$ substrates could subsequently seed bank vole brain homogenate PMCA reactions (**S3 Fig**).

### Mutagenesis and expression of HEK 293 PrP constructs

MoPrP on pcDNA 3.1 was generated by excising MoPrP from pcDNA5/FRT. BV PrP M109 pcDNA 3.1[23] and MoPrP pcDNA 5/FRT were cut using ApaI and HindIII (New England Biolabs, Ipswich, MA, USA). MoPrP was then ligated into the pcDNA 3.1 backbone. Chimeric constructs were generated using site directed mutagenesis using either the GeneTailor Site Directed Mutagenesis System (Invitrogen, Carlsbad, CA, USA) or GeneArt Site Directed Mutagenesis System (Invitrogen)(Table 1). All constructs were confirmed by sequencing. Constructs were then transfected into HEK 293 Freestyle-F cells (Invitrogen, Carlsbad, CA, USA) [55]. After 48 hr, 320 mL of cells were harvested in 80 mL aliquots by spinning at 500 x *g* for 5 min and stored at -80˚C until purification.

**Table 1. Primers and template plasmids used to generate chimeric BV/Mo PrP plasmids.**

| CONSTRUCT | TEMPLATE PLASMID | PRIMERS |
|---|---|---|
| BV(DR) | BV PrP pcDNA 3.1 | **F:** AGTCCCAGGCCTACTACGATGGGAGAAGATCCCGCGCCGTGCTGCTCTTCTC<br>**R:** TCGTAGTAGGCCTGGGACTCCTTCTGATACTG |
| BV(S170DR) | BV(DR) | **F:** CCGGTGGACCAGTACAGCAACCAGAACAACT<br>**R:** TGTACTGGTCCACCGGCCGGTAGTACACTT |
| BV(YS170DR) | BV(S170DR) | **F:** TACTACCGTGAAAACATGTACCGGTACCCTAA<br>**R:** CATGTTTTCACGGTAGTAGCGGTCCTCC |
| BV(LYS170DR) | BV(YS170DR) | **F:** CAGTAAGCCAAAAACCAACCTGAAGCATGTGG<br>**R:** GTTGGTTTTTGGCTTACTGGGCTTGTTCCACT |
| Mo(ES) | Mo PrP pcDNA 5 | **F:** TCAGGCCTATTACGAGGGGAGAAGCTCCAGCAGC<br>**R:** GCTGCTGGAGCTTCTCCCCTCGTAATAGGCCTGA |
| Mo(N155N170ES) | Mo(N170ES) | **F:** ACCGTGAAAACATGAACCGATATCCTAACCAAGTG<br>**R:** CACTTGGTTAGGATATCGGTTCATGTTTTCACGGT |
| Mo(M109N155N170ES) | Mo(N155N170ES) | **F:** CCAAAAACCAACATGAAGCACGTGGCAGG<br>**R:** CCTGCCACGTGCTTCATGTTGGTTTTTGG |
| BV(R) | BV PrP pcDNA 3.1 | **F:** TACTACGAAGGGAGAAGATCCCGCGCCGTGCT<br>**R:** AGCACGGCGCGGGATCTTCTCCCTTCGTAGTA |
| BV(D) | BV PrP pcDNA 3.1 | **F:** CCCAGGCCTACTACGACGGGAGAAGTTCCC<br>**R:** GGGAACTTCTCCCGTCGTAGTAGGCCTGGG |
| BV(S170) | BV PrP pcDNA 3.1 | **F:** GTGGACCAGTACAGCAACCAGAACAAC<br>**R:** GTTGTTCTGGTTGCTGTACTGGTCCAC |
| BV(Y) | BV PrP pcDNA 3.1 | **F:** ACCGTGAAAACATGTACCGGTACCCTAACC<br>**R:** GGTTAGGGTACCGGTACATGTTTTCACGGT |
| BV(L) | BV PrP pcDNA 3.1 | **F:** GCCAAAAACCAACCTGAAGCATGTGGCAG<br>**R:** CTGCCACATGCTTCAGGTTGGTTTTTGGC |
| Mo(S) | Mo PrP pcDNA 5 | **F:** TTACGACGGGAGAAGCTCCAGCAGCAC<br>**R:** GTGCTGCTGGAGCTTCTCCCGTCGTAA |
| Mo(E) | Mo PrP pcDNA 5 | **F:** CAGGCCTATTACGAGGGGAGAAGATCCAGCA<br>**R:** TGCTGGATCTTCTCCCCTCGTAATAGGCCTG |
| Mo(N170) | Mo PrP pcDNA 5 | **F:** GTGGATCAGTACAATAACCAGAACAACTTCG<br>**R:** CGAAGTTGTTCTGGTTATTGTACTGATCCAC |

(*Continued*)

**Table 1.** (Continued)

| CONSTRUCT | TEMPLATE PLASMID | PRIMERS |
|---|---|---|
| Mo(N155) | Mo PrP pcDNA 5 | **F:** ACCGTGAAAACATGAACCGATATCCTAACCAAGTG<br>**R:** CACTTGGTTAGGATATCGGTTCATGTTTTCACGGT |
| Mo(M109) | Mo PrP pcDNA 5 | **F:** CCAAAAACCAACATGAAGCACGTGGCAGG<br>**R:** CCTGCCACGTGCTTCATGTTGGTTTTTGG |
| BV(LDR) | BV(DR) | **F:** CAGTAAGCCAAAAACCAACCTGAAGCATGTGG<br>**R:** GTTGGTTTTTGGCTTACTGGGCTTGTTCCACT |
| BV(YDR) | BV(DR) | **F:** TACTACCGTGAAAACATGTACCGGTACCCTAA<br>**R:** CATGTTTTCACGGTAGTAGCGGTCCTCC |

## Purification of HEK 293 expressed PrP constructs

Pellets thawed on ice were re-suspended in 20 mL of lysis buffer [20 mM MOPS pH 7.0, 0.15 M NaCl, 1% Triton X-100, 1% DOC, Complete™ mini EDTA-free protease inhibitor (Roche, Basel, Switzerland)]. The mixture was solubilized on ice for 30 min, then centrifuged at 100,000 x *g* for 35 min. The solubilized supernatant was applied to a 2 mL IMAC copper sulfate column made with chelating Sepharose (GE Healthcare, Chicago, USA) that was pre-equilibrated with equilibration and wash buffer [20 mM MOPS pH 7.0, 0.15 M NaCl, 10 mM imidazole, 1% Triton X-100]. The column was then washed with 20 mL of equilibration and wash buffer. The sample was eluted with 8 mL of elution buffer [20 mM MOPS pH 7.5, 0.15 M NaCl, 0.15 M imidazole pH 7.0, 1% Triton X-100]. The eluate was diluted 1:1 with pre-SP buffer [20 mM MES pH 5.4, 0.15 M imidazole pH 7.0, 0.15 M NaCl, and 1% Triton X-100]. Next, the eluate was applied over a 2 mL SP Sepharose fast flow (Sigma-Aldrich, St. Louis, MO, USA) exchange column pre-equilibrated with SP wash buffer [20 mM MOPS pH 7.0, 250 mM NaCl, and 1% Triton X-100]. The column was washed with 10 mL SP was buffer and eluted with 8 mL SP elution buffer [20 mM MOPS pH 7.5, 500 mM NaCl, and 1% Triton X-100]. The eluate was then partially deglycosylated by adding 80 µL of Glycerol Free PNGase F (New England Biolabs, Ipswich, MA, USA) to 8 mL of eluate. The mixture was incubated for 24 hr at 37˚C with shaking at 350 r.p.m. The partially deglycosylated substrate was then concentrated and repurified by applying over a 400 µL copper column pre-equilibrated with equilibration and wash buffer. The column was washed with 20 mL of equilibration and wash buffer and eluted with 2 mL of IMAC-CuSO4-elution buffer [20 mM MES pH 6.4, 0.15 M imidazole pH 7.0, 150 mM NaCl, and 1% Triton X-100]. The eluate was loaded into a 3500 MWCO Slide-A-Lyser (Thermo Fisher, Waltham, MA, USA) and dialyzed overnight into dialysis buffer [44].

To ensure that substitutions near the C-terminus of PrP$^C$ do not interfere with addition of the C-terminal GPI-anchor, we treated chimeric PrP$^C$ molecules with either single or double C-terminal substitutions with PI-PLC (**S4 Fig**). It has been previously reported that PI-PLC treatment of GPI-anchored proteins such as PrP$^C$ cause the treated proteins to bind less well to PVDF membranes and migrate more slowly than untreated controls [56]. Accordingly, we observed that all of the HEK-expressed PrP$^C$ chimeras tested, as well as HEK-expressed BV PrP$^C$ native brain PrP$^C$ immunopurified from mouse brain, showed displayed slower migration and lower intensity on western blot after PI-PLC treatment (**S4 Fig**). In contrast,

bacterially-expressed recPrP, which lacks a GPI-anchor, is apparently unaffected by PI-PLC treatment (**S4 Fig**). Thus, all of the chimeric PrP$^C$ substrates used in this study appear to have proper post-translational modifications (**S1 Fig and S4 Fig**) and folding (as judged by solubility in non-denaturing solution and normal behavior during IMAC and ion exchange chromatography. These observations indicate that substitution of homologous residues between mouse and bank vole PrP$^C$ does not cause aberrant metabolism or trafficking of chimeric molecules in HEK 293 cells.

## Supporting information

**S1 Fig. Expression and partial enzymatic deglycosylation of PrPC in FreestyleTM HEK 293 cells.** Western blots showing PrP partially purified from cell lysates expressing the indicated protein, or no protein, before and after partial enzymatic deglycosylation with PNGase F (+ PF), as indicated. **(A) Wild-type constructs. (B) Chimeric constructs.**
(TIF)

**S2 Fig. Effect of partial enzymatic deglycosylation of PrPC substrate on its ability to propagate mouse prions.** Western blots showing three-round reconstituted sPMCA reactions using partially purified Mo PrP$^C$ substrate supplemented with PrP$^{0/0}$ BH and seeded with mouse RML. Substrates were either untreated or partially deglycosylated with PNGase F during purification, as indicated. -PK = samples not subjected to proteinase K digestion; all other samples were proteolyzed.
(TIF)

**S3 Fig. Brain homogenate PMCA reactions seeded with chimeric PrPSc molecules.** Western blots showing PMCA reactions using crude bank vole brain homogenate substrate seeded with various HEK-expressed PrP$^{Sc}$ molecules, as indicated. The chimeric PrP$^{Sc}$ molecules are final round products of 3-round sPMCA reactions using HEK-expressed PrP$^C$ substrates originally seeded with Mo protein-only recPrP$^{Sc}$ or buffer, as indicated.
(TIF)

**S4 Fig. PI-PLC treatment of purified PrP substrates.** Western blots of various purified PrP substrates (bacterially-expressed recPrP, immunopurified native brain PrP$^C$, different HEK-expressed PrP$^C$ chimeras with substitutions at residues 227 and/or 230 as indicated, or HEK expressed bank vole PrP$^C$) treated either with (+) or without (-) 0.25U/mL Phosphoinositide phospholipase C (PI-PLC) for 14 hr at 37 $^o$C with shaking at 800 r.p.m. as indicated. Proteins were transferred from a 12.5% polyacrylamide gel onto a Millipore Immobilon-P PVDF membrane by semi-dry electroblotting, and probed with mAb 27/33.
(TIF)

## Acknowledgments

We thank Emilie Shipman and Jason McLellan for much-appreciated assistance with Freestyle™ cell transfection and culture; Jan Stöhr, Stanley Prusiner, Cathy Chang, and Ta Yuan Chang for providing mouse brains; and Tamutenda Chidawanyika, and Romolo Nonno for valuable advice and discussion.

## Author Contributions

**Conceptualization:** Cassandra M. Burke, Surachai Supattapone.

**Formal analysis:** Surachai Supattapone.

**Funding acquisition:** Surachai Supattapone.

**Investigation:** Cassandra M. Burke, Kenneth M. K. Mark, Daniel J. Walsh, Geoffrey P. Noble, Alexander D. Steele.

**Methodology:** Cassandra M. Burke, Kenneth M. K. Mark, Daniel J. Walsh.

**Project administration:** Cassandra M. Burke, Surachai Supattapone.

**Resources:** Daniel J. Walsh, Abigail B. Diack, Jean C. Manson, Joel C. Watts, Surachai Supattapone.

**Supervision:** Surachai Supattapone.

**Writing – original draft:** Cassandra M. Burke, Surachai Supattapone.

**Writing – review & editing:** Cassandra M. Burke, Kenneth M. K. Mark, Daniel J. Walsh, Abigail B. Diack, Joel C. Watts, Surachai Supattapone.

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
