## [Decision Letter · Decision Letter 0]

28 Apr 2020

Dear Dr. Suppatapone,

Thank you very much for submitting your manuscript "Identification of a Homology-Independent Linchpin Domain Controlling Prion Protein Conversion" for consideration at PLOS Pathogens. As with all papers reviewed by the journal, your manuscript was reviewed by members of the editorial board and by several independent reviewers. In light of the reviews (below this email), we would like to invite the resubmission of a significantly-revised version that takes into account the reviewers' comments.

In particular, we would like you to include data addressing concerns raised by reviewers 1 and 3 about characterization and amount of chimeric PrP substrates expressed in HEK293 cells, as aberrant processing may affect conversion efficiency. Furthermore, both reviewers strongly recommend to include robust quantification of western blot signals and statistical analysis when conversion efficiencies are compared (e.g. Fig. 3B and Fig. 7).

We cannot make any decision about publication until we have seen the revised manuscript and your response to the reviewers' comments. Your revised manuscript is also likely to be sent to reviewers for further evaluation.

Sincerely,

Sabine Gilch, PhD

Pearls Editor

PLOS Pathogens

Michael Malim

Section Editor

PLOS Pathogens

Kasturi Haldar

Editor-in-Chief

PLOS Pathogens

orcid.org/0000-0001-5065-158X

Michael Malim

Editor-in-Chief

PLOS Pathogens

orcid.org/0000-0002-7699-2064

Reviewer's Responses to Questions

**Part I - Summary**

Reviewer #1: The manuscript by Burke et al. uses an in vitro conversion system to study the conversion of normal cellular prion protein (PrPC) from mice (Mo) or bank voles (BV) by a protein only form of recombinant mouse PrPSc (recPrPSc). They show that recPrPSc converts bank vole, but not mouse, PrPC to PrPSc. They further show that BV amino acid residues 227 and 230 are needed for recPrPSc to convert PrPC as is lack of glycosylation at the second glycosylation site (G2) in PrPC. They conclude that residues 227, 230 and a lack of glycosylation at G2 comprise a C-terminal domain that acts as a “linchpin” for conversion of PrPC to PrPSc.

This is a relatively straightforward study looking at the ability of an artificially made mouse prion, protein only recPrPSc, to convert mouse PrPC. While the data and design are appropriate, it’s not clear that an artificially generated prion which does not convert its native substrate to PrPSc is a valid system to study the conversion mechanism of natural prions. The argument that the inability of recPrPSc to convert mouse PrPC enables the study of important conformational determinants without taking into consideration amino acid sequence does not make much logical sense. The primary sequence of a protein is intimately tied to conformation, and earlier studies have already made the point that conformational differences driven by amino acid sequence significantly impact PrPSc formation.

Reviewer #2: In this study, Burke et al utilized a previously developed yet unique in vitro PrPSc amplification system to identify the extreme C-terminus of Bank Vole PrPC, namely amino acid residues E227 and S230, as the linchpin for prion conversion, which is a significant advance in our understanding of prion protein misfolding and formation of PrPSc. The results are solid and the manuscript is well written.

Reviewer #3: Burke et al investigated the molecular PrP domains involved in the conversion of bank vole PrPC by prions formed from recombinant PrP of mouse origin (Mo-recPrPSc). By studying the convertibility of chimeric mouse/bank vole PrP by Mo-recPrPSc seeds in PMCA reactions, the authors identified the extreme C-terminal region of PrPC as an important domain controlling PrP conversion.

The conclusions are partly supported by the data.

**Part II – Major Issues: Key Experiments Required for Acceptance**

Reviewer #1: 1) All of the Mo-BV chimeric constructs are expressed in HEK 293 cells. However, no data are shown to indicate whether or not the various chimeras are processed normally, i.e. fully glycosylated, GPI anchored, and expressed on the cell surface. Aberrant processing of the chimeric PrPC molecules, especially those with mutations at 227 and 230 in the C-terminus where the GPI anchor is attached, could affect their ability to convert to PrPSc. The authors should provide information on whether or not the chimeras tested were appropriately processed by the cell.

2) Based on the data in Fig. 3B, the author states that the BV(DR) substrate with two amino acid changes converts less well than the BV(LDR) and BV(YDR) substrates with 3 amino acid substitutions (compare row 2 to rows 6 and 7). However, this is not clear from the data shown in Figure 3 since the signal intensities on the gels shown are very similar. If possible, the authors should include quantitation data to show this more convincingly.

3) In Figure 7, recPrPSc does not convert the BV(DR) construct to PrPSc. However, in Figure 3 it does convert BV(DR), albeit at a lower efficiency. This suggests some variability in conversion efficiency between experiments that the authors should address.

4) In Figure 7, conversion of mouse (ES) by hamster, sheep, and deer PrPSc brain homogenate yields conversion products in the first 2 rounds that are protease-resistant but do not appear to decrease in size. By the third round, there does appear to be some protease-resistant, truncated product in the sheep and deer reactions. Do the authors have any explanation for why Mo(ES) appears to be fully protease resistant through two rounds of sPMCA only to become partially protease-resistant by the third round?

5) A mouse PrPC substrate where the two glycosylation sites have been mutated yielding an unglycosylated product (G3) is not converted to PrPSc by recPrPSc. How does this result impact interpretation of the other experiments in the paper, all of which use partially purified, PNGaseF treated, deglycosylated PrPC substrate? If fully glycosylated PrPC substrate is used, how does that effect the ability of protein only recPrPSc to convert the substrate to PrPSc?

Reviewer #2: Prion conversion is a very complex process and many factors contribute to it, and the main influencer is PrP sequence. The current study identifies E227 and S230 as the key amino acids that make Bank Vole PrPC the almost universal substrate for prion conversion by a variety of prion strains, including the in vitro formed protein-only recombinant PrPSc. Although the substitutions of D227 and R230 of mouse PrPC to E227 and S230, respectively, render the mouse chimeric PrPC susceptibility to various prion strains, which strongly supports the key role of E227 and S230, it requires more evidence to conclude such domain as the “general” linchpin for prion conversion, since the only two PrPC species investigated are Bank Vole and mouse. It would be much more convincing if hamster PrPC with E227 and D230 mutations could also be converted by RML, Scrapie and CWD.

Reviewer #3: The conclusions are partly supported by the data:

1. Because all the study is based on the difference of PrPSc levels in the PMCA products over serial rounds, the manuscript should include robust quantification and statistical analysis of PrPSc when mutation effects are compared.

This is especially important as i) there is no indication on the reproducibility of the experiments shown, ii) PrPSc abundance is generally low in the western blots (compare for instance Fig3 panel B, 2nd vs 3rd lane to measure the effect of the S170 substitution, a key residue of the discussion), iii) PrPres electrophoretic pattern and contribution of initial PrPSc seed to the PrPSc signal detected in the first round are difficult to estimate.

2. The PrPres electrophoretic patterns are difficult to identify. The pattern is frequently polluted by bands of low Mr. Sometimes PK digestion are insufficient, resulting in PrPres patterns migrating like PrPC (see Fig 7 last lane for instance). The pattern is not consistent among the experiments, compare for instance RML seed with Bank vole PrPC fin Fig7B first lane and Fig 2A first lane.

3. The authors expressed chimeric PrPC in human HEK293 cells and purified PrPC before using it as substrate in PMCA experiments:

3.1 The authors do not mention whether these cells express or not human PrPC. Human PrPC could be converted or interfere with chimeric PrP in PMCA reactions.

3.2 Chimeric PrPC is partly deglycosylated with PNGase F (which usually works on denatured material containing SDS?) before use as substrate. Partial deglycosylation may impact the relative convertibility of the chimeric species. In favor of this hypothesis, the authors show that removing one glycan on mouse PrPC facilitates conversion by Mo-recPrPSc, without any other amino acid mutation.

3.3 HEK cells are likely to express variable amounts of PrPC depending on the mutation performed. There is no indication on the concentration of PrPC used in the PMCA reaction. This is a key point as other authors showed that efficient conversion of cell PrPC substrate in PMCA requires concentrated material (PMID: 27384922). In other words, conversion may fail because of insufficient PrP levels, not mutation.

4. The added value of performing experiments on PrPC glycosylation mutants and on �GPI PrPC from transgenics is uncertain. For me, it is a different story.

Regarding �GPI PrPC, another confounding issue is that these mice spontaneously accumulate PrPSc. This may interfere with the PMCA reaction. In addition, PrPC from these mice is lowly glycosylated and this may also impact the conversion by Mo-recPrPSc. Thus ,the relative contribution of the GPI anchor is highly uncertain.

Regarding the second chapter of the discussion (starting with “It is worth noting”) stating that residues E227 and S230 might stabilize PrPC, making it less prone to conversion (do the authors mean self-conversion or templated conversion?). How do the authors reconciliate this idea with the observation that I109 polymorphism is sufficient for spontaneous bank vole PrPC conversion into prions?

**Part III – Minor Issues: Editorial and Data Presentation Modifications**

Reviewer #1: The figure legend to Figure 7 describes panel C but this panel is not in the figure shown. This should be corrected.

Reviewer #2: Minor points:

1. One of the limitations of this study, as the authors have pointed in the discussion part, is the lack of a way to confirm the infectivity of various chimeric PrPSc produced in the PMCA reactions. The authors argue that “However, we have previously shown that similar reactions containing BV PrPC substrate and seeded with protein-only recPrPSc produces fully infectious prions, making it likely that we are also studying a process relevant to infectious prion formation in this study”. While it is impractical to ask for in vivo bioassays, the author may consider evaluating the seeding ability of various chimeric PrPSc by using BV and mouse brain homogenates containing WT PrPC as substrates in PMCA reactions, which could serve as the prion infectivity surrogate.

2. In figure 1, the numbering for all PrP sequences follows the hamster PrPC sequence (23-231). However, in figure 8, the numbering follows mouse PrP sequence (23-230).

3. In figure 5B, Mo(N155N150ES) on the third row should be Mo(N155N170ES).

Reviewer #3: - Please add page and lanes number to facilitate the comments.

- The introduction is conveying incomplete information, particularly for a general audience:

As examples:

- Introduction, 2nd chapter, about the transmission barriers. “Humans are susceptible to CJD…”. This is obvious and not an example of a cross-species transmission. 2nd sentence: “…but not to CWD or scrapie”. The references mentioned here refer to transgenic modelling to study whether human PrP can be converted by CWD or scrapie. Whether humans are in fine susceptible or not, we do not know. They are counter-examples of conversion, see for instance the work published by Cassard et al. (PMID: 25510416 ), same for CWD with studies in primates.

- Resuming the species to a paper from 1999 by S. Priola is a big shortcut. The topics would deserve more than that. Transmission barriers are more than “dictated by amino acid sequence complementarity between the incoming PrPSc seed and PrPC substrate”. There are remarkable examples showing that the strain type is a key determinant too. For example, propagation of human CJD in bank vole without species barrier, efficacy of atypical BSE to propagate in human PrP mice, while classical BSE does not, etc. The description of the transmission barrier is too simplistic.

PLOS authors have the option to publish the peer review history of their article (what does this mean?). If published, this will include your full peer review and any attached files.

Reviewer #1: No

Reviewer #2: No

Reviewer #3: No
---

## [Decision Letter · Decision Letter 1]

1 Aug 2020

Dear Dr. Supattapone,

Thank you very much for submitting a revised version of your manuscript "Identification of a Homology-Independent Linchpin Domain Controlling Mouse and Bank Vole Prion Protein Conversion" for consideration at PLOS Pathogens. As with all papers reviewed by the journal, your manuscript was reviewed by members of the editorial board and by several independent reviewers. All reviewers appreciated the extent of your revisions. We are likely to accept this manuscript for publication, but ask you to address a minor concern raised by reviewer 1. While they were satisfied with removal of the Mo(N170ES) chimera from the data sets, they noted that reference to this constructs needs to be removed from both Table 1 and SFig. 1.

Sincerely,

Sabine Gilch, PhD

Pearls Editor

PLOS Pathogens

Michael Malim

Section Editor

PLOS Pathogens

Kasturi Haldar

Editor-in-Chief

PLOS Pathogens

orcid.org/0000-0001-5065-158X

Michael Malim

Editor-in-Chief

PLOS Pathogens

orcid.org/0000-0002-7699-2064

Reviewer Comments (if any, and for reference):

Reviewer's Responses to Questions

**Part I - Summary**

Reviewer #1: Overall, the authors have satisfactorily addressed the concerns in my original review.

Reviewer #2: The authors addressed my concerns.

Reviewer #3: The revised version of the manuscript by Burke et al. has been significantly improved by providing additional information and results and quantification of the effects of amino acids substitution on the conversion process. The discussion has also been significantly improved.

I have no more specific concerns.

**Part II – Major Issues: Key Experiments Required for Acceptance**

Reviewer #1: None

Reviewer #2: The authors addressed my concerns.

Reviewer #3: (No Response)

**Part III – Minor Issues: Editorial and Data Presentation Modifications**

Reviewer #1: In response to my concern about whether or not the mutant PrP molecules were appropriately processed by the cell, the authors added new supporting information in SFig 1 and SFig 4, and removed the chimera Mo(N170ES) from the data set. This is fine and addresses my concerns, but the chimera Mo(N170ES) is still in both Table 1 and SFig. 1. These references to this construct may need to be removed as well.

Reviewer #2: The authors addressed my concerns.

Reviewer #3: (No Response)

PLOS authors have the option to publish the peer review history of their article (what does this mean?). If published, this will include your full peer review and any attached files.

Reviewer #1: No

Reviewer #2: No

Reviewer #3: No
---

## [Editor Report · Decision Letter 2]

11 Aug 2020

Dear Dr. Supattapone,

We are pleased to inform you that your manuscript 'Identification of a Homology-Independent Linchpin Domain Controlling Mouse and Bank Vole Prion Protein Conversion' has been provisionally accepted for publication in PLOS Pathogens.

Best regards,

Sabine Gilch, PhD

Pearls Editor

PLOS Pathogens

Michael Malim

Section Editor

PLOS Pathogens

Kasturi Haldar

Editor-in-Chief

PLOS Pathogens

orcid.org/0000-0001-5065-158X

Michael Malim

Editor-in-Chief

PLOS Pathogens

orcid.org/0000-0002-7699-2064
---

## [Editor Report · Acceptance letter]

2 Sep 2020

Dear Dr. Supattapone,

We are delighted to inform you that your manuscript, "Identification of a Homology-Independent Linchpin Domain Controlling Mouse and Bank Vole Prion Protein Conversion," has been formally accepted for publication in PLOS Pathogens.

Best regards,

Kasturi Haldar

Editor-in-Chief

PLOS Pathogens

orcid.org/0000-0001-5065-158X

Michael Malim

Editor-in-Chief

PLOS Pathogens

orcid.org/0000-0002-7699-2064